# MQL-Assisted Hard Turning of AISI D2 Steel with Corn Oil: Analysis of Surface Roughness, Tool Wear, and Manufacturing Costs

**Bogdan Arsene [1,\*], Catalin Gheorghe [1], Flavius Aurelian Sarbu [1], Magdalena Barbu [1], Lucian-Ionel Cioca [2,3] and Gavrila Calefariu [1]**

[1] Department of Engineering and Industrial Management, Transilvania University of Brasov, 29 Eroilor Street, 500036 Brasov, Romania; gheorghe.c@unitbv.ro (C.G.); sflavius@unitbv.ro (F.A.S.); magda.n@unitbv.ro (M.B.); gcalefariu@unitbv.ro (G.C.)

[2] Department of Industrial Engineering and Management, Faculty of Engineering, Lucian Blaga University of Sibiu, 550024 Sibiu, Romania; lucian.cioca@ulbsibiu.ro

[3] Academy of Romanian Scientists, 010071 Bucharest, Romania

\* Correspondence: arsene.bogdan@unitbv.ro

**Abstract:** Precision hard turning (HT) gained more and more attention in the cutting industry in the last years due to continuous pressure of the global market for reducing costs, minimizing the environmental and health issues, and achieving a cleaner production. Therefore, dry cutting and minimal quantity lubrication (MQL) became widely used in manufacturing to meet the environmental issues with respect to harmful cutting fluids (CFs). Vegetable oils, in MQL machining, are a promising solutions to petroleum-based CFs; however, the effects and performance on surface roughness and tool wear in HT with ceramic inserts remain unclear. To address this limitation, hardened AIDI D2 steel and pure corn oil, rich in saturated and monounsaturated fatty acids, cheap and widely available, have been used to conduct dry and MQL experiments at different cutting speed and feeds. Results show that corn oil is suitable as cutting lubricant in HT, creating a strong anti-wear and anti-friction lubricating film which improves the roughness with 10–15% and tool life with 15–20%, therefore reducing costs. Best surface roughness values (Ra = 0.151 μm, Rz = 0.887 μm, Rpk = 0.261 μm) were obtained at 180 m/min and 0.1 mm/rev. The analysis of variance shows that corn oil has statistical significance on roughness, validating the results.

**Keywords:** hard turning; MQL; tool wear; roughness; vegetable oil; corn oil; green manufacturing

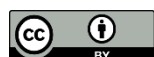

## 1. Introduction

Nowadays, companies from the modern manufacturing industry face the challenges to reduce costs, increase productivity, and quality of products and minimize the environmental issues, in order to remain competitive in a global and dynamic market [1]. The high quality requirements, the continuously demand for high productivity, the global competition, and the trend toward a cleaner production, push the manufacturers to continuously develop innovative strategies in machining, improve the production processes, take advantage from economic opportunities, and create a greener workplace and eco-friendly machining [2]. To enhance these aspects, the machining process should be as economic as possible, providing a high material removal rate, high quality, reliability, and flexibility. Traditionally, grinding is the conventional method or the dominant process in the finishing of high hardened parts. However, the innovations from the last two decades in terms of machine-tools rigidity and accuracy and cutting inserts manufacturing, materials, grades, geometries, and coatings has led to an increase in the use of hard turning

(HT) instead grinding [3]. HT became a widely spread cutting technology for both roughing and finishing of revolution parts made by hardened steel with a hardness higher than 55 HRC, in many industries, e.g., automotive, bearings, dies, gears, and shafts, due to the numerous advantages it takes compared to grinding from an economic, technological, and environmental point of view. Viewed as a sustainable and attractive alternative to grinding, more environmentally and human-friendly manufacturing processes such as precision hard turning are able to yield a high quality machined surface and offer many benefits to manufactures in the context of higher flexibility, material removal rate, lower setup and cycle time, and lower costs [4]. Surface quality in finishing operation is recognized as main indicator for accepting parts; therefore, reducing values of roughness in an economic way is an important task [5,6]. In order to have stability and efficiency in a hard turning process, special cutting inserts are used. Therefore, the materials meeting the tough requirements of hard turning are ceramics and cubic boron nitride (CBN) [7]. Ceramics, having the major advantage against CBN in terms of cost, are known for they high hardness, high wear resistance, high hot hardness, excellent chemical stability, and low mechanical shock resistance [8]. To increase their fracture toughness, other materials, e.g., titanium nitride (TiN), titanium carbide (TiC), or titanium carbonitride (TiCN), are added in their composition, resulting in mixed ceramics [9]. Since the economical aspect of the manufacturing cannot be ignored, the tool cost is of high interest. Therefore, tool life in terms of tool wear must be regarded [10]. The strength of the parts, heat generated in cutting zone, high cutting forces, and low thermal conductivity causes difficulties in machinability and lead to severe wear and surface integrity issues [11]. Severe tool wear leads to low quality, failure, scrap, and breakdown and therefore should be controlled and reduced in order to maintain a stable and reliable process and enhance productivity [12]. The control of wear, the reduction of friction, of temperature in the cutting zone and cutting forces is achieved through cutting fluids (CFs) [13]. Doubtless, the CFs are improving tool life and surface quality [14]. However, CFs, also known metal working fluids (MWFs), represent over 10% of production costs [15,16]. Being an essential constituent for improving machinability, CFs ensure cooling, lubrications, and process stability and provide lower tool wear, better surface integrity, and higher accuracy [17,18]. However, even if the MWFs are necessary and indispensable in many machining sectors, the environmental, costs, and health concerns will restrict this method in the future by regulations and laws [19,20]. Since the early 1990s, considerable efforts have been made to completely eliminate MWFs and many green cutting lubrication techniques have been proposed, e.g., dry cutting, self-lubricating tools, or MQL [20,21]. The MQL technique is a cooling and lubricating technique which consists of spraying a small amount of oil directly in the cutting zone at a flow rate between 10 and 200 milliliter/hour (mL/h) [17,21]. Practically, it uses atomized micro-droplets that can penetrate the cutting zone more effectively to achieve lubrication [20,22]. The technique merges the sustainability with productivity growth and was proposed as a viable alternative to flood cooling, as a solution to environment issues and professional risks, and has shown over time many advantages, e.g., lower costs with CFs, energy, tools, and equipment, flexibility in machine relocation, a healthier workplace, higher quality of products, and productivity growth [10,23]. Meanwhile, MQL gained more and more attention and quickly became a global sustainable solution. Nevertheless, conventional CFs are mineral oil based and the substitution is mandatory. Having better biodegradable and toxicity properties than petroleum or chemical-based oils which cause severe pollution, the vegetable oils gained popularity and have vast potential for many application as green cutting fluid (GCF) [24,25]. Vegetable oils are a feasible and promising alternative to mineral, synthetic, and semi-synthetic oils based on MWF, which are harmful for both worker and environment. Vegetable oils, due to their environmental friendly characteristics, e.g., low toxicity, biodegradability, and renewability, and proprieties useful in cutting, e.g., high flash point and viscosity, excellent lubrication capacity, low volatility, relatively low cost, have ecologic and economic benefits and cleaner and sustainable manufacturing. In addition, vegetable oils create a thick strong

film which cools the machining zone, reduce friction coefficient, cutting forces, tool wear, and heat generation [26,27]. Since many differences in physicochemical properties of different vegetable oils exist [20], the state of the art present a variety of research involving vegetable oils in machining, e.g., cotton seed [25,28,29], coconut [30–37], soybean [31,33,38,39], canola [31,36,37,40], castor [41–44], palm [43,45–47], groundnut oil [33,43], and sunflower [39,42,44].

The wide use of vegetable oils and associated benefits led to new approaches in manufacturing as green manufacturing. This is an essential component for sustainability [17,48]. According to Maruthi and Rashmi [49], green manufacturing "is a philosophy rather than a standard or a process" which aims to "minimizes waste and pollution through product and process design" with the main objective of sustainability [49]. Sustainability becomes an important strategy for the business environment, which drives innovation, aiming at energy and resource savings, alignment with legal regulations and customer requirements and gaining competitive advantage [50]. Therefore, green manufacturing is beneficial not only for the environment but also for the business environment [17,50].

### 1.1. Approach toward Green Manufacturing

Green manufacturing refers to methods and techniques for reducing costs and environmental impact [16], for creating safe and clean work-places [51], for reducing harmful emissions, for minimizing and eliminating wasteful consumption of resources, and to recycle as much as possible [17,49]. In manufacturing, the cutting processes are resource-intensive and therefore concerns and future research in green processing should focus on the main elements leading to the implementation of green manufacturing, such as energy efficient machine tools, recyclable cutting tools, environmentally friendly and biodegradable lubricants, and renewable energy sources [52]. Green cutting is becoming increasingly popular due to environmental safety and occupational health concerns [53]. It involves an approach that does not use the traditional flood cooling. Classic cutting fluids contain chemicals that are dangerous and harmful for the human body, such as biocides (used to control the growth of microorganisms and bacteria) [54,55], chlorine, sulphur, and phosphorus [56], which can cause irritation of skin and eyes, odors and others [51]. The traditional lubricating method used in cutting, i.e., flood cooling with classical CFs, is harmful both for the environment and employee health [57]. These liquids can cause health problems and important environmental damage. Adverse health effects can occur mainly due to the evaporation of liquids produced by high temperatures in the cutting area, producing aerosols with a particle size between 0.1 and 10 microns (μm). Inhalation of toxic elements stored in these liquids can cause serious problems in the eyes, nose, and airways. Vegetable oils are biodegradable in nature and generally have a higher molecular weight than mineral oils, which gives them superior lubricating properties. Most of the vegetable oils consist primarily of triacylglycerides which have molecular structure with three long chain fatty acids [9,20]. This structure and the high content of saturated and monounsaturated fatty acids provide a high strength and dense lubricant film that interact strongly with metallic surfaces, reducing wear and friction and making them suitable for MQL [20,34]. The film is formed due to physical and chemical adsorptions of the vegetable oil molecules and metal surface [28]. Vegetable oils have the status of "environmentally friendly" in terms of biodegradability, renewability, and efficiency in cutting operations [18], are much less toxic to the environment and humans [15,58], and lead to decreasing of costs for waste treatment due to high biodegradability [58]. They also have excellent lubricating and biodegradation properties, good temperature-viscosity index, low volatility [59], and can bring significant process improvements compared to mineral oils [15]. These oils have been found as a promising alternative, as CFs, to mineral or synthetic oils and emulsions due to their environmental characteristics and are used by many industries for the development of biodegradable lubricants for various applications [24,26]. The research carried out in this paper confirms this statement and brings concrete experimental results in this regard.

*1.2. Brief Review Regarding Use of Vegetable Oils in Machining*

The state of the art in MQL-assisted hard turning presents few researches regarding usage of vegetable oils as cutting fluids. Chinchanikar et al. [30] investigated the effect of cooling medium on surface roughness in hard turning of AISI 52100 (60–62 HRC) with water-based and coconut oil-based CFs. They used a coated carbide inserts and found that coconut oil produced lowers value of arithmetical mean roughness (Ra) specially at a cutting speed (Vc) between 160 and 200 m/min and depth of cut greater than 0.3 mm. At cutting speed lower than 160 m/min, dry turning was superior and the best surface roughness (Ra ≈ 0.5 μm) achieved also in dry. Gunjal and Patil [31] experimentally studied the hard turning of AISI4340 (52–54 HRC) using vegetable oil-based CF under MQL, with coated carbide inserts. Canola oil, coconut oil, and soybean oil were used in experiments at an air pressure of 6 bar and fluid flow rate of 50 mL/h and the results showed that canola oil provided best results in terms of tool wear and tool life at cutting speed of 200 and 220 m/min, while at 240 m/min all three oils led to same tool life. However, the tools life was rather small (3–9 min). Xavior and Adithan [32] analyzed the influence of coconut oil, on surface roughness and tool wear when turning AISI 304 steel with carbide insert, compared with soluble an straight oil and noticed that the coconut oil led to lower values of both. Feed was found to be most influencing factor on surface roughness with about 64% contribution followed by type of fluid and depth of cut with about 14%. Ghuge and Mahalle [33] investigated the MQL (50 mL/h) turning of AISI 4130 with carbide insert, many types of vegetable, i.e., soybean, sunflower, coconut, and groundnut, dry and flood cooling, and reported that vegetable oil-assisted MQL provides lower cutting temperature and better surface roughness compared with dry and flood cooling. The best performance was achieved with soybean oil. Ananda Ghatge et al. [35] studied the performance of coconut and neem oils in comparison with mineral oil in turning of AISI 2205 duplex stainless steels (32 HRC) at different cutting parameters and found that both vegetable oil-based cutting fluids are superior to mineral cutting fluid in terms of surface roughness an temperature. Padmini et al. [36] compared under MQL condition the canola and coconut oil, pure and with nanoparticles, with dry and conventional cutting fluid in turning of AISI 1040 steel (30 HRC). Main cutting force, cutting temperature, tool wear and surface roughness were improved in MQL with vegetable oil; the coconut oil was found superior to canola oil. Padmini et al. [37] investigated the machining performance of pure coconut, sesame oil, and canola oil compared with dry, conventional cutting fluid and nanofluids (fluid with inclusion of nanoparticle, e.g., molybdenum di sulphide, copper, alumina-oxide, or graphene added in a certain percentage), in turning of AISI 1040. The performance of pure oils was lower compared with nanofluids; however, in terms of cutting temperature and main cutting force, better results were obtained when vegetable oils were used compared to dry cutting or with conventional fluid. Raj et al. [38] found that the soybean oil lead to lower cutting forces in MQL turning of AISI 4340 (45 HRC) compare with flood cooling even is vegetable or mineral oil flood cooling. Ghuge and Mahalle [39] analyzed the MQL (50 mL/h) turning of 4130 with uncoated brazed carbide tool and with soybean oil, sunflower oil, and mineral-based cutting fluid and found that the use of vegetable oils have led to lower cutting force and power consumption with 7–9%. Belluco and De Chiffre [40] analyzed the influence of rapeseed oil in drilling austenitic stainless steel 316L compared with commercial mineral oil-based under flood conditions. They used also some formulated oils containing rapeseed oil, ester oil, and meadow foam oil with sulphur and phosphor additives and found that all formulated oils and the simple rapeseed oil produced better results than the mineral reference oil in terms of tool life, thrust force, corner wear, and chip tangling around the tool. Elmunafi et al. [41] conducted experiments to establish the influence of castor oil on surface roughness, forces and tool wear compared with dry turning of AISI 420 stainless steel (46–48 HRC) and shown that working under MQL with 50 mL/h flow rate, 5 bar air pressure and vegetable oil, the values obtained were improved. The Ra values obtained are in range 0.3–0.4 μm. Pereira et al. [42] tried

many types of oils, e.g., high-oleic sunflower, sunflower, castor, and recycled oil, in milling of Inconel 718. From a technical point of view, the high-oleic sun flower oil performed better than canola and simple sunflower oils, under MQL conditions. Suresh et al. [43] performed desirability function analysis (DFA) and technique for order preference by similarity to an ideal solution (TOPSIS) to find the best the optimal process parameter setting. They turned under MQL condition AIDI D3 steel using three types of insert, three levels of cutting speed, feed, depth of cut, and cutting fluid (i.e., castor, palm and groundnut), and response variables measured were surface roughness, material removal rate, and specific energy. The authors found that by DFA the best CF is the groundnut oil and by TOPSIS the best CF is the castor oil. Cutting speed was the most significant parameter (≈46% contribution) followed by feed (≈32% contribution) and depth of cut, while cutting fluid and insert style were almost negligible. Khunt et al. [44] analyzed the drilling of Aluminum-6063 alloy under dry, flood cooling and MQL conditions with castor and sunflower oil. They found that thrust force, torque and surface roughness are improved in MQL compared with the other two methods. Mahadi et al. [45] reported that the boric acid aided in palm oil improve the surface roughness compared with conventional lubricant in MQL-assisted turning of AISI 431 (30 HRC) stainless steel. Rahim and Sasahara [46] experimented the drilling of Inconel 718 with solid carbide drill under flood and MQL conditions, using water soluble CF, synthetic ester and palm oil for MQL. The palm oil it proved to be the best choice in this application since the lower temperature, thrust force, and torque were obtained under MQL with this oil. Same Rahim and Sasahara [47] furthermore investigated the drilling of Ti–6Al–4V in same cooling conditions. Likewise, previous research of palm oil led to the best results, measuring the same characteristics. Last but not least, Gajrani et al. [60] compared under MQL (0.5 MPa air pressure and oil flow rate of 35 mL/h) and flood cooling, a mineral oil and a vegetable oil-based CF in hard turning of AISI H13 (56 HRC) with tungsten carbide cutting insert. They reported that the vegetable oil-based eco-friendly bio-CF provide better results and significantly reduce cutting and feed force, friction coefficient, and surface roughness.

Most of the literature focuses on the study of machining soft materials with different types of vegetable oil. The existing literature slightly analyzed MQL-assisted hard turning of high hardness steels (hardness greater than 50 HRC) with commercial available (additive free) vegetable oil, while the corn oil and ceramic wiper inserts lack from researches. To fill this gap further research should be undergo and this paper aims to investigate these aspects with regard to surface roughness and tool wear and to contribute to the field.

### 1.3. Objectives and Hypotheses

The corn oil is richer in saturated fatty acids than sunflower or canola oil and richer in monounsaturated fatty acids than coconut, sunflower, and soybean; therefore, it makes it appropriate as cutting lubricant. In addition, the price and availability make it more affordable for the global cutting industry. Since it is assumed by the research community [30,33,38,45–47,60–64] that MQL, and in many cases even dry, hard turning provide better responses (surface roughness, tool wear, cutting forces, power, etc.) compared with flood cooling, the paper does not include experiments with a conventional coolant. "The battle" is given by dry and MQL cutting as they are the easiest and cheapest to implement, compared to cryogenic or other hybrid cooling methods.

The aim of this paper is to establish the performance of pure corn oil, which represent a novelty in MQL-assisted hard turning with vegetable oil, with respect to surface roughness and tool wear, compared with dry machining, at variable cutting speed and feed, with wiper ceramic inserts commonly used as cheaper alternative to PCBN, in order to meet the conditions of green manufacturing. The assessment of con oil as potential bio-cutting fluid in HT is achieved through statistical analysis and modeling and tool wear criterion. The article is organized in sections as follows. Section 2 describes the research methodology. Section 3 presents the experimental result, while Section 4

discusses the statistical analysis and modeling, tool wear investigation, and cost analysis. Finally, the last section presents the conclusions of the study and future research directions.

The aim of the paper is achieved through the following specific objectives: determining, based on the literature, the known limits in this field, establishing the research parameters, methodology and equipment, statistical analysis of results for mathematical modelling, and interpretation of the results in order to capitalize them.

The research hypotheses were developed based on the practical experience gained so far and the analysis of the current researches presented in the bibliography. These hypotheses (denoted H) are

**Hypothesis 1 (H1).** *The corn oil in MQL-assisted HT is a significant parameter for responses.*

**Hypothesis 2 (H2).** *The surface finish in MQL-assisted HT is significantly better than in dry cutting, when corn oil is used.*

**Hypothesis 3 (H3).** *Tool wear in MQL-assisted HT is less than in dry cutting when corn oil is used.*

**Hypothesis 4 (H4).** *The goal of green manufacturing, when corn oil is used, is achievable in economic conditions.*

## 2. Materials and Methods

The tests have were out to analyze and evaluate the performance of corn oil as cutting fluid in MQL-assisted hard turning. In addition to this method, dry experiments were conducted for comparison. The machine-tool used was a 5.5 kW (7.5 HP) precision CNC lathe type PLG42, made by Po Ly Gim Machinery Co., Ltd., Tongluo, Taiwan, equipped with the Fanuc 0i Mate-TC control system. Round bars of AISI D2 steel hardened at 55 ± 1 HRC, having a diameter of 50 mm and a length of 63 mm, were used as work piece for tests. All bar specimens were rough turned to avoid deviations and grooved for 3 mm width and 2 mm depth on the middle to ensure a smooth exit of the insert, resulting in 30 mm cutting length. AISI D2 is a cold work high carbon high chromium die steel with high hardness, good dimensional stability, strong hardenability, and corrosion resistance often used for punches, dies, forming rolls, burnishing tools, gauges, knurls, and other tooling. The high hardness and chromium content causes rapid tool wear. Therefore, it makes it suitable for investigation. The hardness was assured by through-hardening and tempering and was measured with a portable hardness tester Sauter HMM, provided by Kern & Sohn, Balingen-Frommern, Germany. The chemical composition of the AISI D2 steel is given in Table 1.

**Table 1.** Chemical composition of AISI D2 (%).

| C | Si | Mn | P | S | Cr | Mo | Ni | V | Cu |
|------|------|------|-------|---------|------|------|------|------|------|
| 1.57 | 0.28 | 0.29 | 0.022 | <0.0003 | 11.4 | 0.74 | 0.23 | 0.74 | 0.08 |

The corn oil is obtained by pressing or extraction and consist of saturated (about 13% palmitic acid) and unsaturated (about 87% oleic and linoleic acid) fatty acids [65]. It is generally cheaper and has a lower viscosity than sunflower, soybean, or canola oils, which are widely used in machining—research and industry. The physical and chemical property characteristics of corn oil are presented in Table 2 [66–68].

**Table 2.** Physical and chemical characteristics of corn oil.

| Physical Characteristics | Unit | Temperature Range | | | | |
|---|---|---|---|---|---|---|
| | | **22 °C** | **35 °C** | **50 °C** | **80 °C** | **160 °C** |
| Dynamic viscosity | [MPa·s] | 53–59 | 31–37 | 22–23 | 11–12 | |
| Specific heat | [kJ/kg·K] | - | 1.673 | - | 1.783 | 2.021 |
| Flashing point | °C | | | 234 | | |
| Acid value | mg KOH/g | | | 0.6 | | |
| Saponification value | mg KOH/g | | | 187–195 | | |
| Relative density (20 °C/water °C) | - | | | 0.917–0.925 | | |

The experiments were performed in two steps. In the first step, the experiments targeted the analysis of the influence of the cooling type on surface roughness using ceramic wiper inserts, while in the second, the experiments aimed the influence of the cooling type on tool wear when hard turning with wiper ceramic. Whereas hard turning requires negative rake angle and main cutting edge prepared with chamfer, hone, or both to face the high forces, negative ISO ceramic insert type CNGA 120412 T01020WG, grade CC650, with corner radius R 1.2 mm and wiper geometry, composed from mixed ceramic and recommended for high speed finishing of grey cast irons and hardened materials, manufactured by Sandvik Coromant, Sandviken, Sweden, was selected. The cutting edge was prepared to produce a chamfer of 0.1 mm × 20°. The tool holder (tool for turning) was type PCLNL 20 × 20, of the same manufacturer. The combination between insert and holder provided a clearance angle $\alpha$ of 6°, orthogonal rake angle $\gamma$ of −6°, inclination angle $\lambda$ of −6°, and a tool cutting edge angle $\kappa$ of 95°. The effective rake angle, $\gamma_{ef}$, resulted in −26°. The cutting parameters were selected based on insert manufacturer's recommendation and own expertise. The cutting speed was set to 120, 150, and 180 m/min, respectively, and the feed rate was set to 0.1, 0.15, and 0.2 mm/rev. The cutting length was 30 mm. The depth of cut was kept fixed for all experiments at 0.1 mm. Table 3 summarize the inputs and their level.

**Table 3.** Input parameters and their levels.

| Input Parameter | Unit | Symbol | Level | | |
|---|---|---|---|---|---|
| Cutting speed | [m/min] | Vc | 120 | 150 | 180 |
| Feed | [mm/rev] | f | 0.1 | 0.15 | 0.2 |
| Lubrication type | - | Lub. | Dry | MQL | |

The MQL system was developed as in [69] and consisted from a gun spray, a flexible arm, pneumatic hose, pressure regulator, and air compressor. The pressure was set at 0.4 MPa (4 bar) and the flow rate to 50 mL/h. The flow rate was adjusted using the flow regulator of the gun spray. The gun spray nozzle was oriented at an inclination of about 30° from the tool rake face, at a distance of 35–40 mm and the spraying direction was adjust in order to cool and lubricate the tool–work piece interface, i.e., on the insert rake face. The experiment set-up is presented in Figure 1a.

A full factorial design of experiment (DOE) was employed, with four experiments for each level, resulting in 72 tests. The tests are equally divided as follow: 36 tests for each type of lubrication, dry and MQL, being further divided into three sets of 12 test according to Vc levels (3 levels). Furthermore, these sets were divided again in three subgroups based on feed levels (3 levels), resulting in four experiments, main experiment and three replicates for confirmation, for each level. The full factorial design permits, all possible combinations of the factors levels, allowing for easy construction and analysis of the design [70]. To establish the significance level of input parameters, the statistical analysis has been performed in terms of main effects plots, analysis of variance (ANOVA), and regression modeling. ANOVA was performed, with the aid of the MINITAB V.19 software,

made by Minitab LLC, State College, PA, USA, to establish the influence and statistical significance of cutting conditions, i.e., feed, cutting speed, and lubrication type on responses, and regression models were accomplished, based on experimental results, for validation. A high level of coefficient of determination ($R^2$) shows the accuracy of the regression models and the significance of experimental results [10].

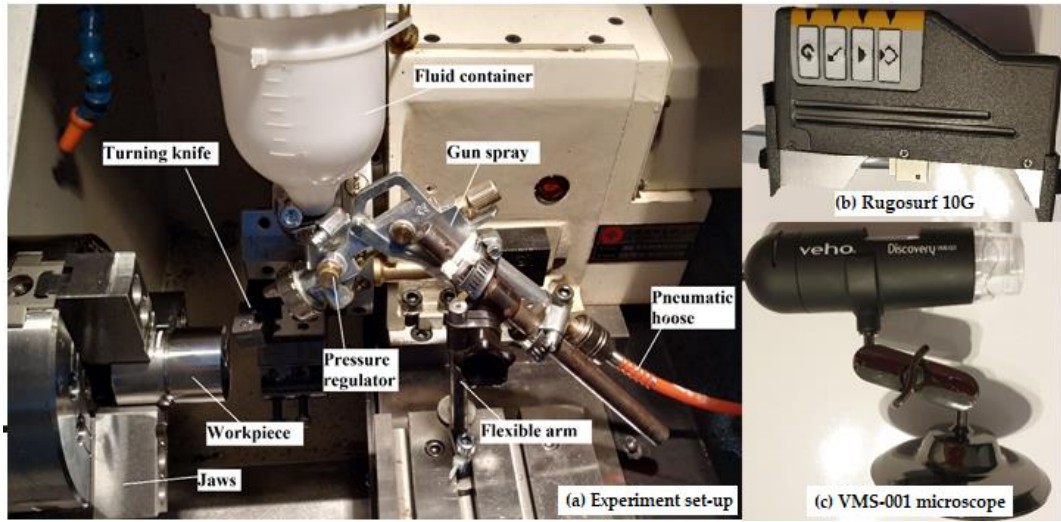

**Figure 1.** Experiment conditions.

The surface roughness was measured with a roughness tester Rugosurf 10G, made by TESA SA, Renens, Switzerland, (Figure 1b), with Gaussian filter, cut-off length $\lambda$c of 0.8 (evaluation length 4 mm). The parts were fixed on a magnetic stand to avoid the displacement and the roughness gauge was fixed to measure on a straight axial line, basis on cylindrical surfaces alignment principles for accurate measurements. Three measurements, after each first test (30 mm cutting length), were made at three different location at about 120° apart, in the middle of the machined surface and the mean value was recorded. For the experiment's validation, the roughness for replicates 2, 3, and 4 was measured once, after each test, in one location, in the middle of the machined surface, and the values were recorded. Part surface finish was evaluated according to ISO 4287-1997 for amplitude parameters and ISO 13565-2 for profile height distribution parameter. Arithmetical mean roughness (Ra), peak to valley height (Rz) and reduced peak height (Rpk) was measured and recorded. Ra is the most popular indicator which define the finishing of a surface and a criterion for technology characterization. Rz is widely used on the surfaces which are considered "important" and Rpk, part of Rk family, became widely used on the applications subject to high loads because the peaks (from the functional surfaces) are the first prone to wear and a high value can lead to premature fail or lower lifetime of the component. In the second step, tool wear was investigated in dry and MQL machining. Same MQL parameters, MQL lubricant, tool holder, and work-piece were chosen for comparison. The tool life was set based on flank wear (Vb) criterion. The inserts were measured until they reached 0.3 mm flank wear. Flank wear was measured with an optical microscope VMS-001 made by Veho, Southampton, UK, (Figure 1c) with 20–200× magnification, 1.3MP, and up to 1280 × 960 resolution, while surface roughness was measured with same roughness gauge as in first trials. The cutting speed and feed was kept constantly for both experiments to 180 m/min, respectively 0.1 mm/rev and 0.1 mm depth of cut.

The cost analysis was performed to identify the opportunity of using MQL instead of dry cutting, economically speaking, and to highlight the economic benefits of using corn oil. The tool and vegetable oil costs were quantified and the return of investment for the lubrication system was determined.

## 3. Results

The test was planned with the aim of relating the influence of input parameters on the surface roughness. Table 4 presents the experimental results of Ra, Rz, and Rpk roughness, for all factors combination (Vc, f, and Lub.) and four replicates, for experiments confirmations. The results shows that the roughness value Ra was obtain in the range of 0.151–0.452 μm, the Rz value in the range of 0.887–2.534 μm, and the Rpk value in the range of 0.261–1.049 μm. The values are similar or better than some from a classic finishing operation, e.g., grinding. Best Ra and Rpk values, of 0.151 and 0.261 μm, were obtained at run order 63, with cutting speed 180 m/min, feed 0.1 mm/rev, and MQL (third replicate), while best Rz value of 0.887 μm was obtained at same cutting condition at run 64 (fourth replicate). The worst values were observed at cutting speed 120 m/min and 0.2 mm/rev feed, i.e., Ra of 0.452 μm (at second replicate), Rz of 2.534 μm, and Rpk of 1.049 μm (at first replicate).

**Table 4.** Experimental results.

| Run | Input Parameters | | | Responses | | |
|---|---|---|---|---|---|---|
| | **Vc** | **f** | **Lub.** | **Ra (μm)** | **Rz (μm)** | **Rpk (μm)** |
| 1 | 120 | 0.1 | Dry | 0.270 | 1.517 | 0.442 |
| 2 | 120 | 0.1 | Dry | 0.250 | 1.420 | 0.457 |
| 3 | 120 | 0.1 | Dry | 0.263 | 1.394 | 0.455 |
| 4 | 120 | 0.1 | Dry | 0.274 | 1.564 | 0.596 |
| 5 | 120 | 0.15 | Dry | 0.256 | 1.462 | 0.429 |
| 6 | 120 | 0.15 | Dry | 0.288 | 1.527 | 0.506 |
| 7 | 120 | 0.15 | Dry | 0.311 | 1.912 | 0.594 |
| 8 | 120 | 0.15 | Dry | 0.336 | 2.056 | 0.689 |
| 9 | 120 | 0.2 | Dry | 0.413 | 2.534 | 1.049 |
| 10 | 120 | 0.2 | Dry | 0.452 | 2.321 | 0.834 |
| 11 | 120 | 0.2 | Dry | 0.399 | 2.157 | 0.693 |
| 12 | 120 | 0.2 | Dry | 0.387 | 2.113 | 0.640 |
| 13 | 150 | 0.1 | Dry | 0.240 | 1.371 | 0.429 |
| 14 | 150 | 0.1 | Dry | 0.256 | 1.540 | 0.471 |
| 15 | 150 | 0.1 | Dry | 0.233 | 1.287 | 0.422 |
| 16 | 150 | 0.1 | Dry | 0.236 | 1.314 | 0.419 |
| 17 | 150 | 0.15 | Dry | 0.282 | 1.729 | 0.478 |
| 18 | 150 | 0.15 | Dry | 0.283 | 1.843 | 0.437 |
| 19 | 150 | 0.15 | Dry | 0.337 | 2.296 | 0.650 |
| 20 | 150 | 0.15 | Dry | 0.295 | 1.897 | 0.634 |
| 21 | 150 | 0.2 | Dry | 0.284 | 1.729 | 0.532 |
| 22 | 150 | 0.2 | Dry | 0.331 | 1.881 | 0.613 |
| 23 | 150 | 0.2 | Dry | 0.291 | 1.673 | 0.572 |
| 24 | 150 | 0.2 | Dry | 0.317 | 1.847 | 0.648 |
| 25 | 180 | 0.1 | Dry | 0.205 | 1.253 | 0.319 |
| 26 | 180 | 0.1 | Dry | 0.185 | 1.073 | 0.294 |
| 27 | 180 | 0.1 | Dry | 0.194 | 1.183 | 0.321 |
| 28 | 180 | 0.1 | Dry | 0.189 | 1.044 | 0.314 |
| 29 | 180 | 0.15 | Dry | 0.239 | 1.359 | 0.372 |
| 30 | 180 | 0.15 | Dry | 0.210 | 1.189 | 0.371 |
| 31 | 180 | 0.15 | Dry | 0.271 | 1.723 | 0.629 |
| 32 | 180 | 0.15 | Dry | 0.270 | 1.643 | 0.619 |
| 33 | 180 | 0.2 | Dry | 0.398 | 1.830 | 0.791 |
| 34 | 180 | 0.2 | Dry | 0.403 | 1.887 | 0.593 |

| 35 | 180 | 0.2 | Dry | 0.383 | 1.774 | 0.389 |
| 36 | 180 | 0.2 | Dry | 0.407 | 1.895 | 0.563 |
| 37 | 120 | 0.1 | MQL | 0.245 | 1.451 | 0.439 |
| 38 | 120 | 0.1 | MQL | 0.211 | 1.215 | 0.337 |
| 39 | 120 | 0.1 | MQL | 0.232 | 1.295 | 0.455 |
| 40 | 120 | 0.1 | MQL | 0.233 | 1.409 | 0.362 |
| 41 | 120 | 0.15 | MQL | 0.257 | 1.578 | 0.438 |
| 42 | 120 | 0.15 | MQL | 0.286 | 1.453 | 0.439 |
| 43 | 120 | 0.15 | MQL | 0.253 | 1.516 | 0.423 |
| 44 | 120 | 0.15 | MQL | 0.250 | 1.778 | 0.373 |
| 45 | 120 | 0.2 | MQL | 0.306 | 1.530 | 0.520 |
| 46 | 120 | 0.2 | MQL | 0.315 | 1.646 | 0.533 |
| 47 | 120 | 0.2 | MQL | 0.311 | 1.635 | 0.548 |
| 48 | 120 | 0.2 | MQL | 0.382 | 2.302 | 0.750 |
| 49 | 150 | 0.1 | MQL | 0.233 | 1.345 | 0.404 |
| 50 | 150 | 0.1 | MQL | 0.213 | 1.239 | 0.411 |
| 51 | 150 | 0.1 | MQL | 0.227 | 1.376 | 0.418 |
| 52 | 150 | 0.1 | MQL | 0.215 | 1.221 | 0.425 |
| 53 | 150 | 0.15 | MQL | 0.208 | 1.240 | 0.314 |
| 54 | 150 | 0.15 | MQL | 0.236 | 1.222 | 0.403 |
| 55 | 150 | 0.15 | MQL | 0.251 | 1.285 | 0.407 |
| 56 | 150 | 0.15 | MQL | 0.268 | 1.337 | 0.444 |
| 57 | 150 | 0.2 | MQL | 0.266 | 1.573 | 0.478 |
| 58 | 150 | 0.2 | MQL | 0.269 | 1.488 | 0.502 |
| 59 | 150 | 0.2 | MQL | 0.277 | 1.604 | 0.502 |
| 60 | 150 | 0.2 | MQL | 0.271 | 1.563 | 0.400 |
| 61 | 180 | 0.1 | MQL | 0.154 | 0.959 | 0.261 |
| 62 | 180 | 0.1 | MQL | 0.163 | 0.976 | 0.296 |
| 63 | 180 | 0.1 | MQL | 0.151 | 1.005 | 0.261 |
| 64 | 180 | 0.1 | MQL | 0.154 | 0.887 | 0.280 |
| 65 | 180 | 0.15 | MQL | 0.210 | 1.311 | 0.321 |
| 66 | 180 | 0.15 | MQL | 0.192 | 1.169 | 0.318 |
| 67 | 180 | 0.15 | MQL | 0.237 | 1.421 | 0.391 |
| 68 | 180 | 0.15 | MQL | 0.233 | 1.386 | 0.379 |
| 69 | 180 | 0.2 | MQL | 0.350 | 2.073 | 0.655 |
| 70 | 180 | 0.2 | MQL | 0.378 | 2.191 | 0.623 |
| 71 | 180 | 0.2 | MQL | 0.330 | 2.043 | 0.460 |
| 72 | 180 | 0.2 | MQL | 0.359 | 2.112 | 0.550 |

## 4. Discussion

The ability of the oil to achieve the tool–chip and tool–workpiece interface establishes the effectiveness of lubrications [29]. Compared with positive rake angle insert, the lack of chip breaker and the smooth surface of negative ceramic insert improve the lubricant sliding, the flow rate and the penetration into tool–workpiece interface, therefore increasing wear and friction resisting and improving surface roughness. Anti-friction and anti-wear effects are achieved due to vegetable oil molecules that adhere on the metal surface, resulting in physical and chemical adsorption. Due to the chemical activity of the vegetable oil polar groups, a layered molecular grid is formed on the surface, reducing the friction accordingly. The thin oil layer has high anti-friction, anti-wear and load-carrying capacities reducing therefore the friction and cutting forces [20,29].

### 4.1. Input Parameters Influence on Surface Roughness

Analysis of variance (ANOVA) for Ra, Rz, and Rpk was performed, with MINITAB V.19, in order to identify which parameter significantly influences the surface roughness.

The analysis was carried out at a 0.05 significance level, which means 95% confidence level. Tables 5–7 show the results of the ANOVA. The source of variation with a p-value of less than 0.05 is considered to have statistical significance to the response. The second last column of the tables show the *p*-value and in all cases is smaller than 0.05 (significance level) which means that all factors, i.e., cutting speed, feed, and lubrication method, have a significant influence on surface roughness.

**Table 5.** Results of ANOVA for Ra.

| Source | DF | SS | MS | F | p | Remark |
|--------|----|----|----|----|----|--------|
| Vc | 2 | 0.02202 | 0.011010 | 9.18 | 0.000 | Significant |
| f | 2 | 0.20085 | 0.100423 | 83.72 | 0.000 | Significant |
| Lub. | 1 | 0.03175 | 0.031752 | 26.47 | 0.000 | Significant |
| Error | 66 | 0.07916 | 0.001199 | | | |
| Total | 71 | 0.33378 | | | | |

**Table 6.** Results of ANOVA for Rz.

| Source | DF | SS | MS | F | p | Remark |
|--------|----|----|----|----|----|--------|
| Vc | 2 | 0.6462 | 0.32309 | 6.38 | 0.003 | Significant |
| f | 2 | 4.7342 | 2.36710 | 46.71 | 0.000 | Significant |
| Lub. | 1 | 0.7614 | 0.76143 | 15.03 | 0.000 | Significant |
| Error | 66 | 3.3443 | 0.05067 | | | |
| Total | 71 | 9.4862 | | | | |

**Table 7.** Results of ANOVA for Rpk.

| Source | DF | SS | MS | F | p | Remark |
|--------|----|----|----|----|----|--------|
| Vc | 2 | 0.1528 | 0.076419 | 8.33 | 0.001 | Significant |
| f | 2 | 0.5682 | 0.284121 | 30.98 | 0.000 | Significant |
| Lub. | 1 | 0.1892 | 0.189181 | 20.63 | 0.000 | Significant |
| Error | 66 | 0.6053 | 0.009171 | | | |
| Total | 76 | 1.5155 | | | | |

All three parameters are statistically significant for surface roughness. Feed is the most important parameter which influences the surface roughness. Cutting speed and lubrication method have also statistical significance. Thus, Hypothesis 1(H1) is confirmed, namely, the corn oil in MQL-assisted HT is a significant parameter for surface roughness. The results are in accordance with the objectives initially set and confirm that the MQL-assisted hard turning with vegetable oils improve the surface finish compared with dry cutting. The correlation between input parameters and responses is set by regression analysis. The regression functions for Ra, Rz, and Rpk, for each set of categorical predictors level (dry and MQL), are given by Equations (1)–(6):

- Model summary for Ra: R-sq (square) 98.45%, R-sq adjusted (adj.) 98.3%, and R-sq predicted (pred.) 98.15%:

$$\text{Dry: Ra} = 0.02100 + 0.00375 \times Vc - 0.77 \times f - 0.000018 \times Vc \times Vc + 3.74 \times f \times f + 0.00633 \times Vc \times f, \tag{1}$$

$$\text{MQL: Ra} = -0.02100 + 0.00375 \times Vc - 0.77 \times f - 0.000018 \times Vc \times Vc + 3.74 \times f \times f + 0.00633 \times Vc \times f. \tag{2}$$

- Model summary for Rz: R-sq 98.16%, R-sq (adj.) 97.99%, and R-sq (pred.) 97.81%:

$$\text{Dry: Rz} = 0.1028 + 0.01446 \times Vc + 4.32 \times f - 0.000074 \times Vc \times Vc - 6.9 \times f \times f + 0.0274 \times Vc \times f, \tag{3}$$

$$\text{MQL: } Rz = -0.1028 + 0.01446 \times Vc + 4.32 \times f - 0.000074 \times Vc \times Vc - 6.9 \times f \times f + 0.0274 \times Vc \times f. \tag{4}$$

- Model summary for Rpk: R-sq 96.58%, R-sq (adj.) 96.27%, and R-sq (pred.) 95.86%:

$$\text{Dry: } Rpk = 0.0513 + 0.00595 \times Vc + 0.14 \times f - 0.000025 \times Vc \times Vc + 7.72 \times f \times f - 0.0019 \times Vc \times f, \tag{5}$$

$$\text{MQL: } Rpk = -0.0513 + 0.00595 \times Vc + 0.14 \times f - 0.000025 \times Vc \times Vc + 7.72 \times f \times f - 0.0019 \times Vc \times f. \tag{6}$$

The regression equating established indicates high value of coefficients of determination for all roughness parameters, meaning that the models are statistical significant and highly accurate.

Figure 2 shows the main effect plots for Ra, Rz, and Rpk. The plots indicate that the increase of cutting speed lead to better roughness, especially in Rz and Rpk, the increase of feed to worst roughness in all cases and the use of vegetable oil-assisted MQL improve the surface roughness. As is known in theory, the feed is most important parameter affecting the roughness; the increase of feed significantly affect the finish of the surface. The second parameter seems to be the lubricating method, while cutting speed is less important in this equation. An important conclusion which can be draw from main effect plot is that corn oil assisted MQL lead to surface finish improvements. Therefore, Hypothesis 2 (H2) is confirmed.

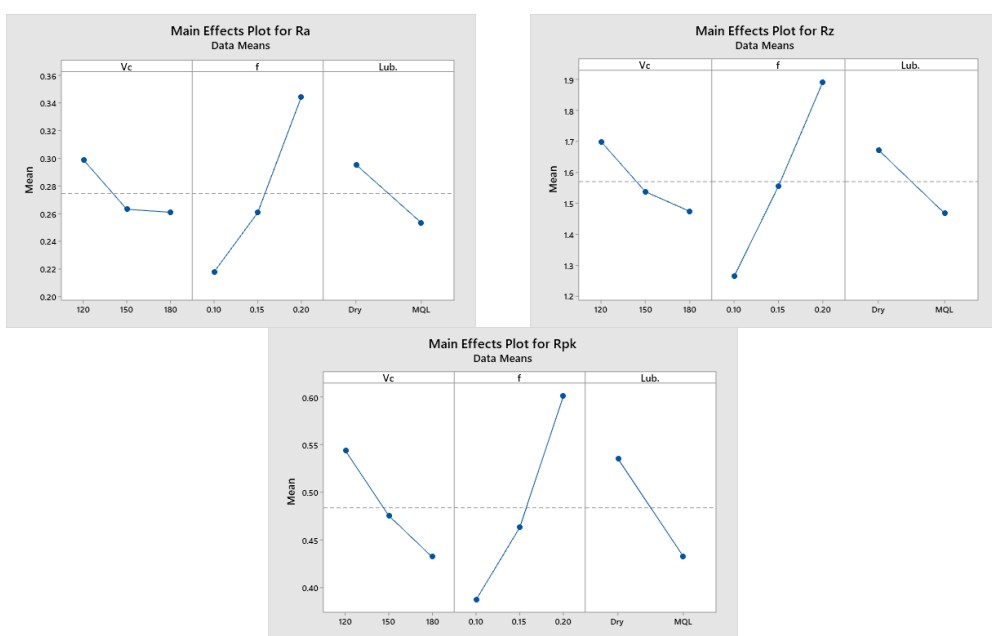

**Figure 2.** Main effect plots for Ra, Rz, and Rpk.

3D plots for each surface roughness, both in dry and MQL cutting, were made for a better understand of the experiment, with the STATISTICA V.8 software (Statsoft, Hamburg, Germany). The plots are presented in Figures 3–5, as mean of the replicates. The best surface finishing was achieved when 0.1 mm/rev feed was set, both for dry and MQL. The rise of feed leads to worst surface for all type of roughness.

In case of Ra, the feed increase negatively influenced the surface finish especially when the cutting speed was set at 120 (low) and 180 (high) m/min. At 150 m/min cutting speed (medium), the change of feed did not affect in the same manner the roughness. This behavior is similar in dry and MQL HT, but compared with dry, the presence of fluid improves the surface finish. We can conclude that intermediate cutting speed (150 m/min)

and high feed (0.2 mm/rev) can be an option for high productivity and acceptable roughness (0.3–0.35 Ra) with better results when corn oil is applied. The increase of speed at low feed (0.1 mm/rev) improves the surface finish from 0.264 to 0.241 and 0.193 μm, in average, in dry cutting and from 0.230 to 0.222 and 0.156, μm in average, in MQL. When oil is applied, the Ra value can effectively be improved by 13%, 8%, and 19% at constant cutting speed. In dry, the Ra can be effectively improved by 27%, while in MQL by 32% when cutting speed is increased. The results suggest that the increase of cutting speed in HT with ceramic inserts improve the surface roughness; this observation has also been identified by other authors [59,71]. When feed is raised, at constant cutting speed, the Ra value increases in all cases. Applying oil, the Ra value can be effectively reduced by 11% to 20% compared to dry. The excellent anti-friction properties of corn oil-assisted MQL is confirmed.

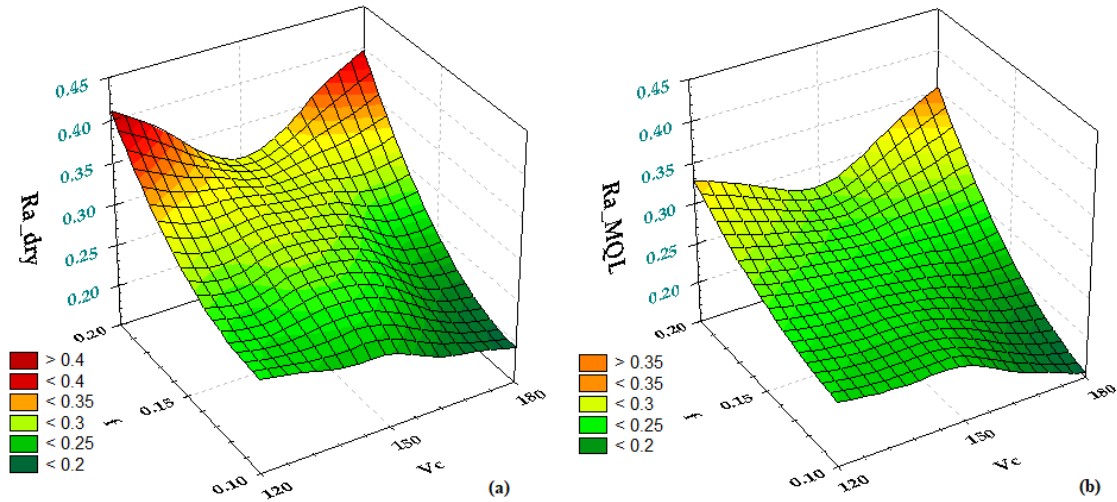

**Figure 3.** Mean Ra vs. feed and cutting speed in dry (**a**) and MQL (**b**).

Best Rz values were obtain at low feed (0.1 mm/rev). When dry cutting, the Rz value was improved once with the cutting speed increase (120 to 180 m/min), from 1.474 to 1.378 and 1.138 μm, mean values, at low feed. Same was observed in MQL, where the Rz value improved from 1.343 la 1.295 and 0.957 μm, mean values. When oil is applied, the Rz value can be effectively improved by 9%, 6%, and 16%. In dry, the Rz can be effectively improved by 23%, while in MQL by 29% when cutting speed is increased.

At high feed (0.2 mm/rev), the trend is similar to Ra. When the cutting speed is raised from 150 to 180 m/min, the surface smoothness decreases both in dry and MQL. This fact can be attributed to the intense cutting parameters combinations which does not favor the achievement of a superior surface quality. Same as Ra, the excellent anti-friction properties of corn oil-assisted MQL is confirmed.

The feed increase, negatively influenced, in a higher manner, the surface finish at 120 and 180 cutting speed, both in dry and MQL. At 150 Vc, the trend is similar to Ra; therefore, same parameters are suitable for Rz in order to achieve high productivity and acceptable roughness (1.6–1.8 Rz) with better results when corn oil is applied. Rz in this case of hard turning with ceramic wipers inserts, is about six time Ra (Rz ≈ 6 × Ra).

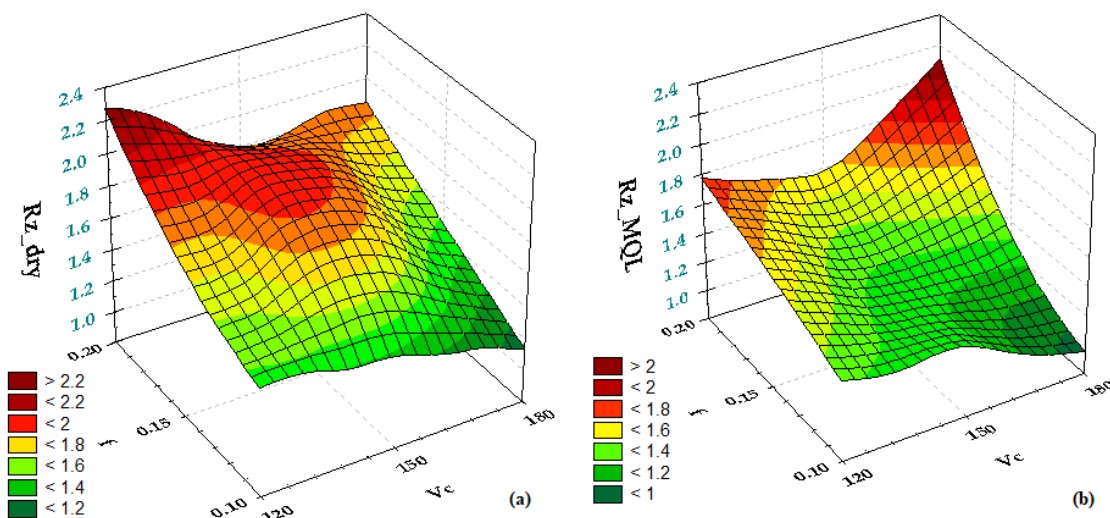

**Figure 4.** Mean Rz vs. feed and cutting speed in dry (**a**) and MQL (**b**).

The best Rpk values were obtain, same as Ra and Rz, at higher speed and lower feed. The values were improved from 0.488 to 0.435 and 0.312 μm, mean values, in dry at low feed. Under condition of MQL, Rpk values slightly increased from 0.398 to 0.415 μm, but improved from 0.415 la 0.274 μm when feed were increased from 0.15 to 0.2 mm/rev. When oil is applied the Rpk value can effectively be improved by 18%, 5%, and 12%, at constant speed. In dry, the Rpk can effectively be improved by 36%, while in MQL by 34% when cutting speed is increased. At high speed, the increase of feed lead to a linear increase of Rpk, while at low cutting speed the increase of feed is exponential and lead to a high increase of Rpk, especially when feed is raised from 0.15 to 0.2 mm/rev. We can conclude that Rpk values are more stable at high speed. Since this parameters is more sensible than Ra and Rz further research is necessary in order to better understand the influence of cutting parameters, but even then we can assume that the influence is similar to Ra and Rz. Same as Ra and Rz, the excellent anti-friction properties of corn oil assisted MQL is confirmed through the resulting values.

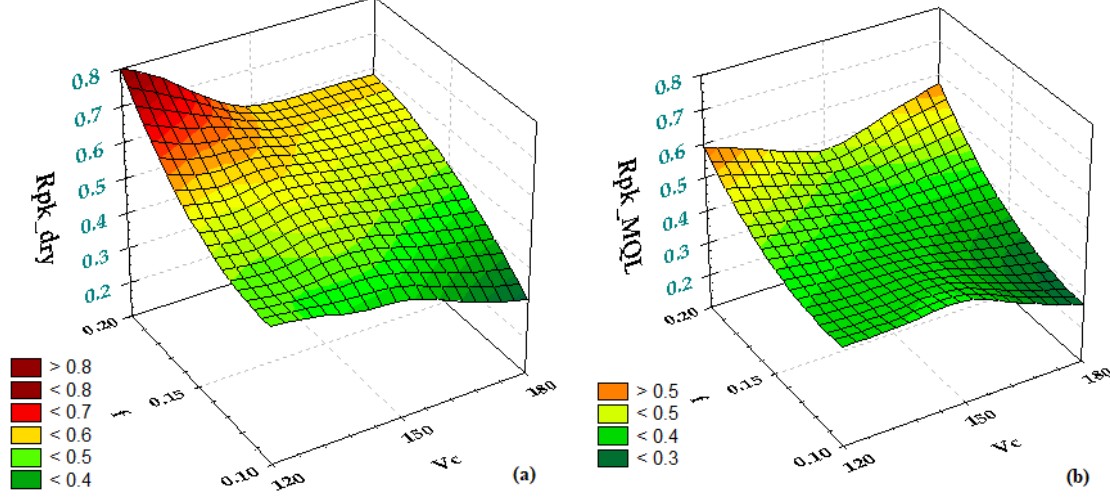

**Figure 5.** Mean Rpk vs. feed and cutting speed in dry (**a**) and MQL (**b**).

Due to the ability of the oil to achieve the tool–chip–workpiece interface in MQL, anti-friction properties, oil molecules that adhere on the metal surface, and relative high saturated and monounsaturated fatty acids, the corn oil is able to create a strong lubricating film in cutting area with high lubricating properties. Therefore, corn oil assisted MQL

hard turning with wiper inserts improve the surface roughness, due to the lubricating effect of oil and his capability to penetrate the tool–workpiece interface. An increase of feed in the range of 0.1–0.15 mm/rev can be easily sustained in order to achieve better productivity without affecting the surface finish excessively. The highest cutting speed provided the best surface roughness and also the best productivity.

### 4.2. Tool Wear Analysis

In cutting operations, due to friction, a large amount of heat is generated in the cutting zone. Most of this, about 80%, is removed through chip and the rest by tool and workpiece (depending on the thermal conductivity) which means that surface alterations, accuracy deviations and rapid tool wear can appear. Decreasing the quantity of heat or improving the transfer into chip leads to better surface finish, longer tool life, and process control. MQL aims to reduce the friction, cutting temperature, heat generation, and to improve the heat transfer through its lubricating capacity, ability to remove the chip from cutting zone and wettability [17,21,72]. Sending air-oil mist under high pressure into the cutting zone, proper cooling, and lubrication is obtained.

Flank wear is the main type of wear which can be found in hard turning. Since ceramic inserts were used in this experiment, the crater wear is not evident due their excellent chemical stability which provide resistance to diffusion [8]. The wear observed in experiments is mainly flank wear generated due to abrasion mechanism caused by the hard particles from workpiece. Figure 6 shows the progression of tools wear with cutting time, in minutes, and the related Ra roughness. In the first minutes, the surface roughness is much better in MQL than in dry cutting, an improve of 30%, while as wear increases the difference between dry and MQL is decreasing to 8% after 16.6 min and 9% after 28 min. Due to excellent anti-friction and anti-wear properties of corn oil and strong lubricating film created tool wear was minimized and surface finish was improved throughout the experiment. The tool life in dry was around 25.5 min while in MQL reached 30 min, at a flank wear of 0.3 mm. The tool life can effectively be improved by 17.6%.

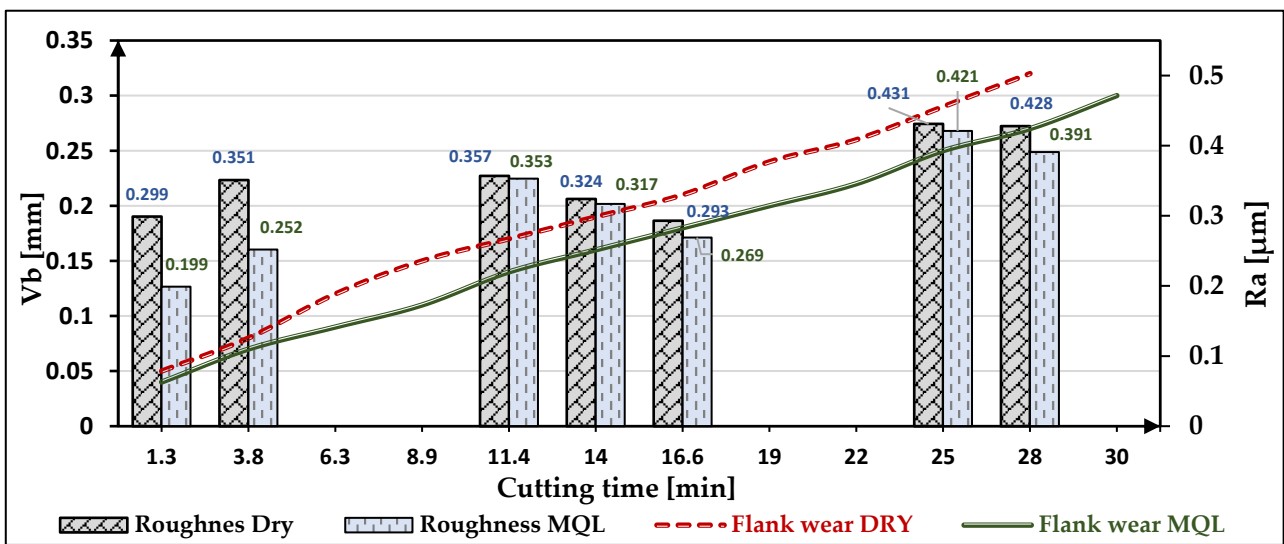

**Figure 6.** Tool wear progression and related Ra.

The main wear mechanism found in experiment is abrasion. In Figure 7, the flank wear can be observed. After 14 min of cutting, the flank wear measured in dry cutting was 0.19 mm, while in MQL was 0.16 mm. No premature fail as chipping or fracturing was observed in this point. Applying corn oil in MQL hard turning of AISI D2, the tool life is improved with 15.8% after 14 min. of cutting.

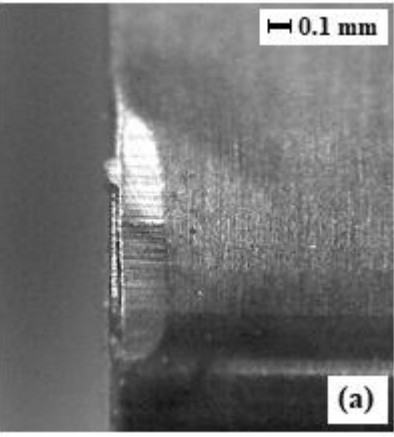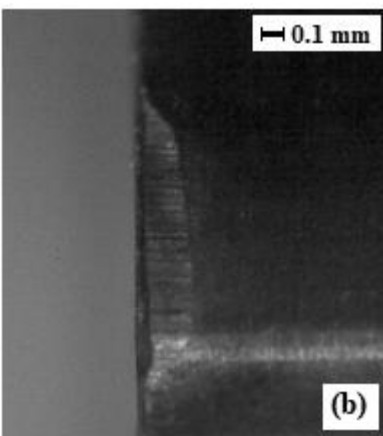

**Figure 7.** Flank wear in dry (**a**) and MQL (**b**) after 14 min cutting time.

Figures 8 and 9 shows the tool wear after 28 min in dry and MQL cutting. When MQL is applied, the flank wear is regular, small abrasive particles are found on the rake face and a small chipping of the edge can be observed. Likewise, due to anti-wear and lubricating and cooling effect of the oil and ability of oil–air mist to eliminate the micro-particles the abrasion of tool when oil is applied is less than in dry. The oil film can be a barrier for the hard micro-particles from the piece which causes the abrasions of the flank face. The Ra obtained after 28 min is 0.391 μm. In dry cutting, severe chipping and adhesion can be observed after 28 min, while the Ra obtained was 0.428 μm. The lack of cooling and lubrications lead to higher quantity of heat, higher temperature, which promotes the diffusion and adhesion. Figure 9 shows, in addition to abrasion, the presence of material adhered on rake face, large chipping, and a more pronounced crater wear due to more quantity of heat generated during cutting.

It can be concluded that the heat generation in dry and lack of lubrications affected the cutting edge in negative way, while in MQL, the properties of corn oil applied on the rake face improved the lubrication and the cooling in tool–chip interface, maintaining a good quality of cutting edge and surface.

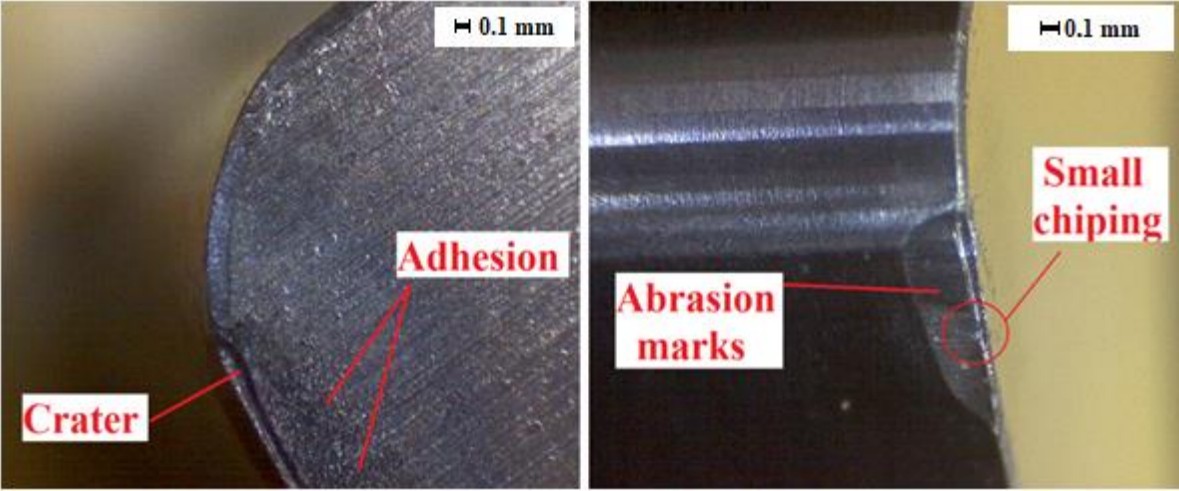

**Figure 8.** Tool wear after 28 min cutting time in MQL.

The Hypothesis 3 (H3) is confirmed, namely, tool wear in corn oil-assisted MQL is less than in dry cutting.

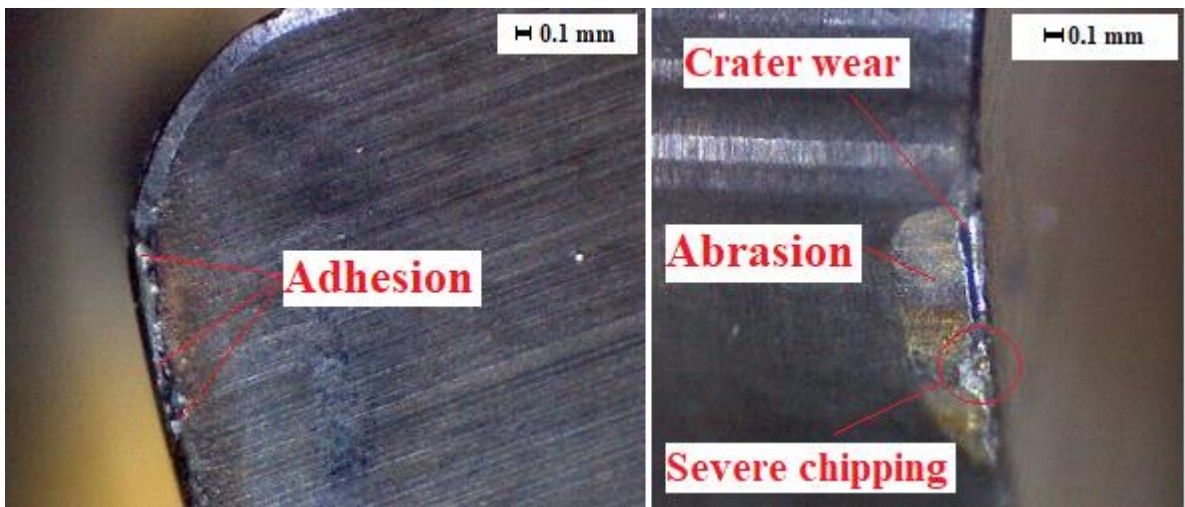

**Figure 9.** Tool wear after 28 min cutting time in dry.

Ceramic inserts are a viable alternative to more expensive CBN insert for finishing hard turning. Since after 28 min of cutting, the surface shows a good quality, i.e., roughness Ra 0.4 μm, the implementation and industrial utilization in hard turning is possible, in order to replace grinding.

### 4.3. Costs Analysis

The use of corn oil as cutting fluid instead of dry turning has also economic benefits, in addition to quality benefits. The lubricating action leads to reduction of friction and cutting forces; lower cutting forces means lower power and lower energy consumption. Many studies proved that cutting forces are reduced in MQL compared with dry cutting [33,41,42,60,73], thus less energy in consumed in machining.

The acquisition price for a ceramic wiper insert is around 15 Eur/insert, as a mean value of four cutting insert available on the market. An increase with 10–20% (in average 15%) in tool life means that the tools cost decrease with 1.5–3 Eur (in average 2.25 Eur) on insert. The energy consumption is smaller in MQL due to lower cutting forces, thus an energy cost saving arise. Since in this research the energy consumed in MQL and dry cutting was not considered, we can assume that the reduction in energy consumed due to lower cutting forces in MQL is equal with the energy consumed for compressed air used in MQL. The oil consumption is 50 mL/h at an average price of 1 Eur/L. The material cost for MQL system, on actual Romanian market, is about 117 Eur composed of: gun spray (≈14 Eur), pneumatic hoose (≈8 Eur), pressure regulator (≈25 Eur), and magnetic arm (≈70 Eur). The values are determined as a mean value of 3–5 prices existing on the market. The compressed air installation was not considered as cost element since is found in any production unit as it is necessary for cleaning the workplace, for chip removing, or for tool changing if the machine has a pneumatic system. Table 8 presents the cost estimation as a comparison between the two cutting methods. The conclusion regarding the economic aspect of the implementation of corn oil-assisted MQL in hard turning is that consistent cost saving is obtained from tool life improvement, while the MQL system cost is recovered in 109 h of cutting, in actual condition. The Hypothesis 4 (H4) is confirmed, namely, the goal of green manufacturing, when corn oil is used, is achievable in economic conditions.

Every improvement in MQL parameters as volume, flow rate, pressure, nozzle distance, and angle lead to longer tool life, lower cutting forces, and lower costs with energy consumed which result a faster cost recovery rate of the lubrication system.

**Table 8.** Cost estimation as a comparison between cutting methods.

| Cost Element | Price | Saving in MQL | |
|---|---|---|---|
| | | **Eur/Time Unit** | **%** |
| Cutting insert/1 pc | 15 Eur/insert 3.75 Eur/corner | +0.562 Eur/ 30′ cutting time | +15% |
| Machine energy consumption | ≈0.12 Eur/kWh | ≈+0.05 Eur/30′ cutting time | −9.5% |
| Compressed air energy consumption | ≈0.12 Eur/kWh | ≈−0.05 Eur/30′ cutting time | +9.5% |
| Oil consumption-50 mL/h | 1 Eur/Liter | −0.025 Eur/30′ cutting time | - |
| Process cost saving | | 0.537/30′ cutting time ≈1.074 Eur/h of cutting | - |

Some of the qualitative benefit us using MQL could also be translated into quantitative benefits. The surface finish improvement in MQL allow to increase feed, with a certain percentage, compared with dry cutting. The increase of feed in MQL will result in same surface finish, same cutting forces, and energy consumption as in dry, but the machining time will decrease. Thus, the productivity will be improved, especially in mass production. This hypothesis should be verified and could be a future research direction as well a more comprehensive cost analysis.

**5. Conclusions**

The paper conducts research with the requirements of green manufacturing, with favorable consequences both technically and economically. Applying vegetable oil as cutting fluid is the right way to a cleaner and sustainable production. The performance of commercial available corn oil was investigating with regard to surface roughness and tool wear, on AIDI D2 hardened steel with ceramic wiper inserts. The following conclusions can be summarized from the research:

- Ra values lower than 0.2 μm, Rz lower than 1 μm and Rpk lower than 0.4 μm can be easily achieved at 180 m/min and 0.1 mm/rev;
- Surface roughness better than grinding can be obtained with ceramic inserts;
- Feed is the most important parameter affecting the surface roughness, followed by the lubricating method and cutting speed. All three parameters are statistically significant;
- Applying corn oil, Ra is improved by 8% to 19%, Rz by 6% to 16%, and Rpk by 5% to 18% at constant speed;
- The increase of speed in HT with ceramic inserts improve Ra by 27%, Rz by 23% and Rpk by 36% in dry cutting, while applying corn oil Ra is improved by 32%, Rz by 29%, and Rpk by 34%;
- Due to relative high saturated and monounsaturated fatty acids corn oil is able to create a strong lubricating film in cutting area with high lubricating properties which improve the tool wear and surface finish;
- The tool wear is improved when corn oil is applied with 17.6% compared to dry cutting;
- Lower tool wear was observed in MQL than in dry. Main wear mechanism were abrasions. In dry cutting adhesion and severe chipping was observed;
- Tool life reached 25.5 min. in dry while in MQL reached 30 min., at 0.3 mm Vb;
- Corn oil can successfully be used as cutting lubricant in hard turning;
- The goal of green manufacturing, when corn oil is used, is achievable in economic conditions.

Further research shall focus on investigating the influence of corn oil on cutting forces, residual stress, and white layers and on some parameters related to sustainability such as power, energy consumption, and carbon emissions, compared with other types of vegetable oil.

**Author Contributions:** Conceptualization, B.A.; methodology, B.A. and G.C.; software, B.A. and M.B.; validation, L.-I.C. and G.C.; formal analysis, B.A. and C.G.; investigation, B.A.; resources, B.A. and F.A.S.; data curation, B.A. and C.G.; writing—original draft preparation, B.A.; writing—review and editing, B.A. and G.C.; visualization, B.A.; supervision, L.-I.C. and G.C.; project administration, B.A. and C.G.; funding acquisition, B.A. and F.A.S. All authors have read and agreed to the published version of the manuscript.

**Funding:** This research received no external funding.

**Data Availability Statement:** The data presented in this study are available on request from the corresponding author.

**Acknowledgments:** Authors would like to thank Sandvik Coromant Romania for their support in providing the cutting inserts for experiments.

**Conflicts of Interest:** The authors declare no conflict of interest.

**Nomenclature**

| | |
|---|---|
| HT | Hard turning |
| MQL | Minimal quantity lubrication |
| CFs | Cutting fluids |
| GCF | Green cutting fluids |
| MWF | Metal working fluids |
| Vc | Cutting speed |
| f | Feed |
| Lub. | Lubrication type |
| Ra | Arithmetical mean roughness |
| Rz | Peak to valley height |
| Rpk | Reduced peak height |
| ANOVA | Analysis of variance |
| Rpk | Reduced peak height |

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
