# Peer review of "MQL-Assisted Hard Turning of AISI D2 Steel with Corn Oil: Analysis of Surface Roughness, Tool Wear, and Manufacturing Costs"

_metals, doi:10.3390/met11122058_

Round 1

Reviewer 1 Report

In this manuscript (Metals-1491561), the performance of corn oil assisted MQL in hard turning with regard to surface roughness, tool wear and costs, compared with dry machining, at variable cutting speed and feed was investigated. However, based on the following comments, the current manuscript still needs further polishing.

  1. The English is not yet sufficient and should be improved significantly because there are many sentences or words that are not professional. The authors are encouraged to ask professional native speaker for English polishing.
  2. What are the advantages of corn oil compared to other oils? There is no doubt that this is important for the entire article. Please add it to the manuscript.
  3. Each part of cost accounting has been quantified and pointed out, and the percentage value of cost reduction should be clearly given.
  4. The logic of the introduction needs to be increased. You can read to the following paper for details: [1] DOI: 10.1016/j.jmapro.2020.09.044. [2] DOI: 10.1007/s00170-020-04969-9.
  5. The surface roughness value of Figure 10 should be marked.
  6. Figures 1/2/3 should be combined to form your experimental conditions.
  7. Try to divide the conclusion into multiple pieces. This will allow readers to find the key information faster.
  8. The ruler of Figure 11/12/13 should be added.
  9. The research work lacks specific reasoning or in-depth mechanism-based interpretation. Try to connect the discussion of each part properly to draw relevant conclusions. The lubricating and synergistic mechanism of corn oil needs to be further mentioned. You can read to the following paper for details: [1] DOI: 10.1007/s00170-021-08235-4. [2] DOI: 10.1016/j.cja.2020.04.029.
  10. The abstract needs to be improved, such as research implications, bottlenecks, and solutions. The purpose of the abstract is to clearly show the uniqueness and the contributions that the work has made such that people will spend time to read the rest of manuscript and get the paper cited. Therefore, it absolutely is important to show “what’s new.” Please revise the abstract to have clear indication of what new knowledge the manuscript brings.

Author Response

Dear reviewer,

We thank you for the suggestions and comments concerning our manuscript and for your patience in reviewing our work. The comments are all valuable and the recommendations made were of real help in completing and improving the manuscript. We have studied all the comments and have made the corrections. We tried our best to improve the paper and made major changes to the manuscript.

An updated version, based on your suggestions, has been uploaded.

We appreciate the intensive work and hope that the new version will meet the requirements of publishing.

                        Thank you again for your valuable evaluation

Yours faithfully,

Authors

Reviewer 2 Report

Remarks

for the paper titled

MQL assisted hard turning with corn oil: analysis of surface 2 roughness, tool wear and manufacturing costs

Line 48                Instead of „material remov rate” → „material removal rate”

Line 114             Instead of „the that period” → „that period”

The writting of dimensions are not the same

E.g.:

Line 118: mL/hour

Line 245 & Line 348: ml/h

Line 557: mL/h

Recommended to use identical one everywhere.

Other: The pascal (symbol: Pa)

Line 348: Instead of „Mpa” → „MPa”

Line 337:            „The tool holder (turning knife)” → „The tool holder (tool for turning)”

Lines 376 & 377: „Flank wear was measured with an optical micro-376 scope VMS-001 made by Veho (figure 3)”.

Question: what is the accuracy, etc. of this measuring equipment?

Line 381: „Figure 3. Optical microscop.” → „Figure 3. Optical microscope

The unit of feed is: „millimeters per revolution [mm/rev]”

So in Lines 345, 379, 390, 393, 439, 448, 466, 467, 475, 480, 482, 491 and 598:

„mm/rot” → „mm/rev”

Line 404: „in all cases is small than 0.05” → „in all cases is smaller than 0.05”

On Figures 7, 8 & 9 the sign of speed should be corrected fro „Va” to „Vc” (twice, for drawing of dry and for drawing of MQL part as well).

On Figures 11 and 12 the same orientation of the inserts, with same magnification should increase to compare the differences in the values of wears.

On Figures 11, 12 and 13 you shoud put a measuring scale what can be used for estimate the values of the wears.

Further remarks:

  1. In the Abstract it should be emphasized:
  • What is the main aim of the paper?
  • What are the challenges?
  • Why was used corn oil? Is it better than other oils,
  • What are the advantages of its use?
  1. Mainly in the Introduction, but sometime later as well, one statement is supported by many, too many, references. E.g. Line 84: 6 references, Line 438: 10 references, etc. It would have been good to read what the main, essential differences are between the various articles cited.
  2. What kind of program was used when making Figures 7-9?
  3. Logical separation of the Conclusions is required. It could help of better understanding.

Author Response

Dear Reviewer,

We thank you for the suggestions and comments concerning our manuscript and for your patience in reviewing our work. The comments are all valuable and the recommendations made were of real help in completing and improving the manuscript. We have studied all the comments and have made the corrections. We tried our best to improve the paper and made major changes to the manuscript.

An updated version, based on your suggestions, has been uploaded.

We appreciate the intensive work and hope that the new version will meet the requirements of publishing.

                        Thank you again for your valuable evaluation

Yours faithfully,

Authors

Round 2

Reviewer 1 Report

This manuscript has been revised carefully according to the reviewers' comments. Now it could be accepted.

Author Response

Dear reviewer,

We appreciate the intensive work and we thank you for the opportunity to submit the manuscript to Metals.

Yours faithfully,

Authors